# Comparison of Refractive Outcomes Between Small Lenticule Extraction Using a 500 KHz and a 2 MHz Femtosecond Laser

**DOI:** 10.3390/diagnostics15192450

**Published:** 2025-09-25

**Authors:** Jung Wan Kim, Youngsub Eom, Dong Hyun Kim, Ji Won Jeong, Seh Kwang Park, Jong Suk Song

**Affiliations:** 1BGN Jamsil Lotte Tower Eye Clinic, Seoul 05510, Republic of Korea; wany9868@naver.com (J.W.K.);; 2Department of Ophthalmology, Korea University Ansan Hospital, Ansan 15355, Republic of Korea; 3Department of Ophthalmology, Korea University College of Medicine, Seoul 02841, Republic of Korea; amidfree@korea.ac.kr (D.H.K.); crisim@korea.ac.kr (J.S.S.)

**Keywords:** lenticule, extraction, myopia, astigmatism, refraction

## Abstract

**Background/Objectives**: The aim of this study was to compare the efficacy of correcting myopia and myopic astigmatism between small incision lenticule extraction (SMILE) surgery performed using VisuMax 500 and that using VisuMax 800. **Methods**: Patients who underwent myopia correction using either VisuMax 500 (493 eyes of 249 patients) or VisuMax 800 (169 eyes of 85 patients) employing the nomogram of VisuMax 500 were enrolled in this retrospective case–control study. At 2 months after the operation, the percentage of eyes achieving a postoperative uncorrected distance visual acuity (UDVA) of 20/20 and residual refractive error were compared between the two groups. Additionally, the percentage of eyes with refractive astigmatism angle of error within ±15° and the mean absolute angle of error were analyzed. **Results**: In the VisuMax 500 and 800 groups, 99% and 97% of eyes achieved a postoperative UDVA of 20/20, respectively. Postoperative residual astigmatism was similar between the two groups, but residual myopia was significantly lower in the VisuMax 800 group (−0.09 ± 0.50 cylinder diopters [CD]) compared to the VisuMax 500 group (−0.21 ± 0.49 CD; *p*-value = 0.005). Additionally, 84% of eyes in the VisuMax 800 group achieved astigmatism correction within ±15° of the intended meridian, significantly outperforming the 71% in the VisuMax 500 group (*p*-value = 0.002), with a significantly smaller mean absolute angle of error (8.3 ± 12.2° and 14.1 ± 20.1°; *p*-value < 0.001). **Conclusions**: Both VisuMax 500 and VisuMax 800 effectively corrected myopia. However, in terms of the accuracy of astigmatism axis correction, VisuMax 800 demonstrated superior precision compared to VisuMax 500. This study may be limited by perceived bias associated with evaluating two platforms from the same manufacturer.

## 1. Introduction

Since the introduction of the small incision lenticule extraction (SMILE) procedure using the VisuMax 500 device (Carl Zeiss Meditec, Jena, Germany) by Dr. Sekundo in 2007, flapless LASIK surgery has gradually gained popularity in the field of laser vision correction surgery due to its unique and excellent visual correction effects [1,2,3]. Although the term “keratorefractive lenticule extraction” has been proposed as a nonproprietary alternative [4], many studies continue to refer to lenticule extraction procedures performed with the VisuMax device as “SMILE” [5,6,7]. With reported effectiveness in myopia correction and long-term stability, SMILE surgery has been widely performed globally over the past decade [8,9,10,11].

The VisuMax 500 machine, which is widely used for flapless SMILE, has certain limitations. Namely, these include the lack of automated centration and cyclotorsion compensation [12,13]. Manual centration based on the pupil center may result in misalignment of the treatment zone, leading to decentered lenticule formation and reduced precision in lenticule extraction [14,15]. In addition, although some degree of cyclotorsion can be manually adjusted, this approach lacks precision and may cause a mismatch between the intended and actual astigmatic axis, thereby reducing the effectiveness of astigmatism correction [13]. Such decentration and cyclotorsion misalignment can lead to suboptimal refractive outcomes [12,16]. Introduced in 2021, the VisuMax 800 device (Carl Zeiss Meditec) offers several updated features, including a higher femtosecond laser repetition rate (2 MHz vs. 500 kHz in the previous model), as well as enhanced options for centration and cyclotorsion adjustment [17]. These improvements may help reduce procedure time and enhance surgical workflow, particularly in challenging refractive cases [18]. In addition, the option for cyclotorsion adjustment may support improved astigmatism correction, although the direct impact of these features on clinical outcomes remains to be fully elucidated. Recent reports by Dr. Reinstein et al. and Dr. Saad et al. have shown that SMILE performed with the VisuMax 800 provides comparable stability, efficacy, and surgical outcomes to those achieved with the VisuMax 500 [16,18].

In the Republic of Korea, approval from the Korea Food and Drug Administration for VisuMax 800 was obtained in 2023, and our institution began performing SMILE surgeries using VisuMax 800 starting in September 2023. In institutions where the existing VisuMax 500 system is used, SMILE surgeries are conducted using the VisuMax 500 nomogram for VisuMax 800, with adjustments to the nomogram made based on the outcomes. Therefore, this study aims to evaluate the initial clinical outcomes of SMILE surgeries performed with VisuMax 800 using the existing nomogram of VisuMax 500 at a single institution over a six-month period. This study compared the corrective effects for myopia and compound myopic astigmatism between the VisuMax 500 and VisuMax 800 systems when utilizing the same nomogram.

## 2. Materials and Methods

This retrospective case–control study encompassed 662 eyes of 334 consecutive patients who underwent SMILE surgery with VisuMax 500 or VisuMax 800 for the correction of myopia and myopic astigmatism between 1 September 2023 and 22 March 2024 at the BGN Jamsil Lotte Tower Eye Clinic in Seoul, Republic of Korea. The study protocol received approval from the Public Institutional Bioethics Committee, Seoul, Republic of Korea (Approval Number: P01-202405-01-017; Approval date: 14 May 2024). Given the retrospective nature of the study, the requirement for individual informed consent was waived by the Ethical Committee. All research and data collection procedures rigorously adhered to the principles outlined in the Declaration of Helsinki. This investigator-initiated study received no financial support from the device manufacturer. As both devices were from the same company, we acknowledge the potential for perceived bias and address it by comprehensively and transparently reporting outcomes.

### 2.1. Study Population

Inclusion criteria for the study comprised patients demonstrating a normal anterior segment of the eye and exhibiting normal macular and optic nerve architecture as assessed using an ultra-wide-field fundus camera (California; Optos PLC, Dunfermline, Scotland, UK) and Cirrus-5000 OCT (Carl Zeiss Meditec, Inc., Dublin CA, USA). Patients diagnosed with retinal pathologies such as retinal holes, lattice degeneration, or minor peripheral retinal detachments underwent therapeutic intervention with photocoagulation laser therapy. Patients were required to have a 1-week break from non-toric contact lenses and a 2-week break from toric contact lenses before preoperative assessments and surgery. Each eye had to achieve corrected distance visual acuity (CDVA) of at least 20/25, with stable refraction for a minimum of 6 months. Additionally, a postoperative residual stromal bed thickness greater than 280 μm was required. Participants had to be adults over 18 years of age to ensure no further myopic progression and be capable of providing informed consent.

The exclusion criteria included prior ocular surgery or trauma, systemic conditions that could potentially impact ocular health, a history of ocular inflammation or retinal detachment, any corneal pathology such as keratoconus or central scar formation, an endothelial cell count below 2200/mm^2^, and a lenticule thickness representing more than 25% of the total corneal thickness. The zonal size was adjusted to meet these criteria. Patients with postoperative complications such as diffuse lamellar keratitis were also excluded.

### 2.2. Patient Examination

Prior to surgery, all patients received comprehensive ocular examinations, which included slit-lamp biomicroscopy, autorefraction/keratometry (RK-F2 Full Auto Ref-Keratometer; Canon, Tokyo, Japan), manifest refraction with CDVA, funduscopy, optic disc and retinal nerve fiber layer imaging, specular microscopy, and Pachymeter. A single Scheimpflug camera test was conducting using the Pentacam system (Oculus Optikgeräte GmbH, Wetzlar, Germany) for tomography and Cirrus 5000 OCT scans for the macula and disc evaluations.

### 2.3. BGN Nomogram for Myopia Correction in SMILE Surgery

The myopic correction values provided to the SMILE machine were determined based on each patient’s spherical manifest refraction values. Different levels of overcorrection were applied depending on the spherical manifest refraction values (Table 1), while the cylinder manifest refraction values remained unchanged. Overcorrection and undercorrection were also adjusted based on patient age. Specifically, patients aged 18–20 years received an additional −0.1 D of overcorrection to the spherical manifest refraction. No additional correction was made for patients aged 21–34 years. Patients between the ages of 35–40 years received an additional +0.1 D of undercorrection, and those aged 41 and above were adjusted with +0.2 D of undercorrection. This nomogram is used solely within BGN Jamsil Lotte Tower Eye Clinic, based on our clinical experience.

### 2.4. Surgical Technique

All SMILE procedures were conducted by a skilled surgeon (J.W.K.) utilizing a VisuMax 500 or VisuMax 800 femtosecond laser system, following a standardized surgical protocol. A “Small” size suction contact glass interface was consistently employed for all treated eyes, with a target cap thickness of 110 μm for all cases. The optical zone was typically set at 6.5 mm but reduced to 6.0 mm to comply with the 280 μm limit of residual stromal thickness. The incision cut was made at 135 degrees on both eyes with a width of 2.5 mm. For the right eye, the incision was made superotemporally, and for the left eye, it was made superonasally. Suction duration ranged from 10 to 11 s, with pulse duration ranging between 220 and 580 femtoseconds for VisuMax 800 and between 22 and 28 s with the same pulse duration for VisuMax 500. Under the standard clinical settings of the VisuMax 500 used at our institution during the study, the spot distance, track distance, and energy were 4.3 µm, 4.3 µm, and 125 nJ, respectively, resulting in a calculated fluence of 676 m J/cm^2^ using the equation of *F* = *E_p_*/(Δ*x*·Δ*y*), where *E_p_* is the pulse energy (J), Δ*x* is the spot distance (cm), and Δ*y* is the track distance (cm) [19]. Under the standard clinical settings of the VisuMax 800, these parameters were 4.0 µm (spot distance), 3.0 µm (track distance), and 115 nJ (energy), yielding a calculated fluence of 958 m J/cm^2^ [19]. In the VisuMax 800 platform, surgeons were able to align the treatment center to the corneal vertex, whose position relative to the pupil center was measured using preoperative Pentacam (Oculus Optikgeräte GmbH) topography. This planned vertex location was used intraoperatively to guide centration with the CentraLign function, allowing precise adjustment of the treatment center via joystick control. After docking the cornea to the interface, cyclotorsional alignment was performed if needed using the OcuLign system by comparing the horizontal reference axis with the preoperative corneal ink marks, and rotational adjustments were made using the joystick. On the VisuMax 500 platform, centration was performed manually by visually aligning the cornea with the contact interface. If cyclotorsional misalignment was observed after docking, the surgeon adjusted the eye’s rotational position by rotating the contact glass. Re-docking was performed if proper alignment could not be achieved. Following the initiation of suction, the femtosecond laser automatically generated an intrastromal lenticule, which was subsequently dissected from the surrounding stroma using a thin hooked spatula (SIFL dissector & lifter; Angel Medical Systems, Bengaluru, India) and extracted through the peripheral corneal incision using Kelman-Mcpherson fragment forceps (Geuder AG, Heidelberg, Germany). Following the surgery, all patients underwent slit lamp examination, with each procedure meticulously recorded to facilitate documentation of any potential complications.

### 2.5. Postoperative Medication

Patients were prescribed 0.5% moxifloxacin ophthalmic solution (Vigamox; Alcon Laboratories, Inc., Fort Worth, TX, USA) and 0.1% fluorometholone (Furuson; Daewoo Pharmaceutical Co., Ltd., Busan, Republic of Korea) to be administered every 6 h until exhaustion. Artificial tears were recommended for postoperative dryness.

### 2.6. Patient Evaluation

At 2 months postoperatively, autorefraction and keratometry assessments were performed, and monocular UDVA was measured in both the VisuMax 500 and 800 groups. If UDVA reached 20/20, no further vision measurements were taken. Ocular examinations were performed to identify any postoperative complications, and patient complaints were meticulously documented.

### 2.7. Main Outcome Measures

The attempted spherical equivalent (SEQ) used to evaluate the predictability of myopia correction in SMILE surgery was defined as the preoperative manifest refraction SEQ. Surgically induced astigmatism vector was defined as the magnitude of the vectorial difference between postoperative autorefraction astigmatism and preoperative autorefraction astigmatism [20,21,22,23,24]. The target induced astigmatism vector was defined as the magnitude of preoperative manifest refraction astigmatism. The refractive astigmatism angle of error was defined as vector angle difference between surgically induced astigmatism and preoperative manifest refraction astigmatism.

### 2.8. Statistical Analysis

Statistical analyses were performed using the Statistical Package for Social Sciences (SPSS version 20.0; IBM Corp., Armonk, NY, USA). Preoperative parameters, postoperative UDVA, and residual refractive error were compared between the VisuMax 500 and 800 groups using Student’s *t*-test and Fisher’s exact test. The percentage of eyes with a postoperative SEQ within ±0.25 D, ±0.50 D, ±0.75 D, and ±1.00 D, as well as the percentages of eyes with postoperative residual astigmatism within ±0.25 cylinder diopters (CD) and ±0.50 CD, were compared between the two groups using Fisher’s exact test. Scatterplots comparing achieved SEQ to attempted SEQ and surgically induced astigmatism vector to target induced astigmatism vector were plotted and evaluated using linear regression analysis in each group. The percentage of eyes with a refractive astigmatism angle of error within ±15°, and the mean absolute angle of error between the two groups, were compared using Fisher’s exact test, Student’s *t*-test, and analysis of covariance (ANCOVA), with preoperative manifest refraction cylinder as a covariate.

## 3. Results

The VisuMax 500 group included 493 eyes from 249 patients, while the VisuMax 800 group included 169 eyes from 85 patients. The mean age (±SD) in the VisuMax 500 group was 23.2 ± 4.6 years, while that in the VisuMax 800 group was 24.0 ± 5.6 years (*p*-value = 0.226). The VisuMax 500 group contained 158 females (63.5%), and the VisuMax 800 group had 43 females (50.6%) (*p*-value = 0.041). The mean preoperative manifest refraction cylinder in the VisuMax 500 group was −0.94 ± 0.68 CD, significantly lower than −1.30 ± 0.91 CD in the VisuMax 800 group (*p*-value < 0.001). However, the mean preoperative manifest refraction sphere in the VisuMax 500 group (−3.68 ± 1.51 D) was higher than that in the VisuMax 800 group (−3.34 ± 1.54 D) (*p*-value = 0.012). Consequently, there was no significant difference in preoperative manifest refraction SEQ between the two groups (−4.09 ± 1.61 D and −4.00 ± 1.53 D; *p*-value = 0.486). There were no significant differences in preoperative UDVA and CDVA, pachymetry, optical zone, lenticule thickness, or residual bed thickness between the two groups (Table 2).

The mean postoperative UDVA in the VisuMax 500 group was 0.00 ± 0.02 logMAR, which was not significantly different from the VisuMax 800 group (0.01 ± 0.02 logMAR) (*p*-value = 0.260). The proportion of eyes with a postoperative UDVA of 0.0 logMAR (Snellen 20/20) was 99% and 97%, respectively, in the VisuMax 500 and 800 groups (Figure 1).

Postoperative residual astigmatism in the VisuMax 500 group was −0.35 ± 0.30 CD, which was not significantly different from −0.36 ± 0.31 CD in the VisuMax 800 group (*p*-value = 0.736). However, the postoperative SEQ in the VisuMax 500 group was −0.21 ± 0.49 D, significantly higher residual myopia compared to the VisuMax 800 group (−0.09 ± 0.50 D) (*p*-value = 0.005). Postoperative SEQ was within ±0.50 D in 73.4% and within ±1.00 D in 96.1% in the VisuMax 500 group, and within ±0.50 D in 76.9% and within ±1.00 D in 95.9% in the VisuMax 800 group (*p*-value = 0.415 and *p*-value = 0.822, respectively). The percentage of eyes with a postoperative SEQ within ±0.75 D in the VisuMax 800 group (93.5%) was significantly greater than that in the VisuMax 500 group (87.0%; *p*-value = 0.024; Figure 2).

The scatterplot comparing achieved to attempted SEQ demonstrated predictability in the procedure for both the VisuMax 500 (R^2^ = 0.896; Figure 3a) and 800 groups (R^2^ = 0.8812; Figure 3b). In the VisuMax 500 group, the proportions of eyes achieving a postoperative SEQ within ±0.25 D, ±0.50 D, ±0.75 D, and ±1.00 D of the target were 40.2%, 67.5%, 86.6%, and 94.3%, respectively. In the VisuMax 800 group, these proportions were 43.8%, 72.2%, 88.8%, and 94.7%, respectively.

In the VisuMax 500 group, the percentages of eyes with postoperative residual astigmatism within ±0.25 CD and ±0.50 CD were, respectively, 55.7% and 82.3% (Figure 4a); in the VisuMax 800 group, these percentages were 55.0% and 79.9% (Figure 4b), respectively (*p*-value = 0.929 and *p*-value = 0.490).

The scatterplot comparing the surgically induced astigmatism vector to the target induced astigmatism vector indicated that the VisuMax 800 group (R^2^ = 0.8496) showed superior predictability in astigmatism correction compared to the VisuMax 500 group (R^2^ = 0.7013). In the VisuMax 500 group, there was a trend towards undercorrection as the amount of astigmatism increased (β1 = 0.7073). However, the VisuMax 800 group exhibited a more linear and effective correction for larger amounts of astigmatism (β1 = 0.8742) (Figure 5).

Analysis of the refractive astigmatism angle of error revealed that 71% of astigmatic eyes in the VisuMax 500 group achieved astigmatism correction within ±15° of the intended meridian (Figure 6a). In contrast, the VisuMax 800 group demonstrated significantly higher accuracy, with 84% of eyes within this range (*p*-value = 0.002; Figure 6b). The mean absolute angle of error was significantly smaller in the VisuMax 800 group (8.3° ± 12.2°) compared to the VisuMax 500 group (14.1° ± 20.1°; *p*-value < 0.001). After adjusting for preoperative manifest refraction cylinder using ANCOVA, the estimated marginal mean absolute angle of error remained significantly lower in the VisuMax 800 group (9.9°, 95% CI: 7.2–12.6°) than in the VisuMax 500 group (12.4°, 95% CI: 10.8–13.9°; *p*-value < 0.001).

## 4. Discussion

This study compared the corrective effects on myopia and myopic astigmatism by applying the same nomogram to both VisuMax 500 and VisuMax 800. The results indicated that there were no significant differences in postoperative UDVA and residual astigmatism between the surgeries conducted with VisuMax 500 and VisuMax 800. However, the SEQ was significantly closer to emmetropia in the VisuMax 800 group. Additionally, the percentage of eyes with a postoperative SEQ within ±0.75 D in the VisuMax 800 group was significantly greater than that in the VisuMax 500 group. One possible explanation for this finding is the higher repetition rate of the femtosecond laser in the VisuMax 800 (2 MHz), which allows for more rapid and continuous lenticule creation. This facilitates smoother stromal dissection and may reduce intraoperative variability, particularly in eyes with higher refractive errors, where the lenticule volume is greater. Moreover, in low myopia, where thinner lenticules are created, the faster and more refined dissection may contribute to more precise tissue separation and reduced variability in lenticule morphology. These factors together may have contributed to the improved predictability and refractive accuracy observed in the SEQ outcomes of the VisuMax 800 group. These findings suggest that surgeons previously using VisuMax 500 could achieve similar or superior refractive outcomes by applying the same nomogram with VisuMax 800.

A previous research study, which utilized the nomogram for VisuMax 500 to correct compound myopic astigmatism using VisuMax 800, reported effective treatment of compound myopic astigmatism using VisuMax 800 [16]. In the same study, 91% of eyes had a postoperative UDVA of 20/20 or better, and 86% of eyes had a postoperative SEQ within ±0.50 D [16]. Similarly, in this study, 97% of eyes had a postoperative UDVA of 20/20, and 76.9% of eyes showed a postoperative SEQ within ±0.50 D in the VisuMax 800 group. Given that the VisuMax 800 shares the same optics and laser delivery design as VisuMax 500, it is reasonable to expect comparable refractive outcomes from both devices, as shown in this and previous studies.

In this study, despite the preoperative astigmatism being significantly higher in patients who underwent surgery with VisuMax 800 compared to those treated with VisuMax 500, there was no significant difference in postoperative residual astigmatism between the two groups. Additionally, regression analysis revealed that while VisuMax 500 showed a trend of undercorrection as the amount of astigmatism increased, VisuMax 800 effectively corrected even higher degrees of astigmatism. Regarding the accuracy of astigmatism correction on the intended meridian direction, VisuMax 800 demonstrated higher precision compared to VisuMax 500. Consequently, a significantly larger number of astigmatic eyes (84%) in the VisuMax 800 group showed an angle of error within 15°. Similar to the findings of this study, previous research also showed that when correcting compound myopic astigmatism with VisuMax 800, 81% of astigmatic eyes demonstrated an angle of error within 15° post-correction [16]. The SMILE surgery nomogram used at our institution only adjusts the sphere values of the measured manifest refraction, without altering the cylinder values. From this perspective, when compared to VisuMax 500, VisuMax 800 appears to be able to correct astigmatism with greater predictability and accuracy, even in cases of higher corneal astigmatism.

The reduced angle of errors in the astigmatic axis observed in the VisuMax 800 group compared to the VisuMax 500 group can be attributed to differences in the centration and cyclotorsion adjustment methods between the two devices. In VisuMax 500, centration is manually adjusted using the Placido topography ring image and Hirschberg reflex printouts to determine the relative position of the corneal vertex to the pupil border [12,16]. Cyclotorsion adjustment also requires manual reorientation of the immobilized eye by rotating the contact glass. In contrast, VisuMax 800 incorporates advanced methods for centration and cyclotorsion adjustment through its CentraLign and OcuLign software. These programs continuously display the vector difference between the corneal vertex and the treatment cone’s position during the docking process, and after docking, allow for the adjustment of the treatment axis to match prepositioned corneal marks using a reticule guideline [16]. Therefore, it is believed that these integrated software enhancements in the VisuMax 800 system contribute to the superior astigmatic correction outcomes compared to the manual techniques used in VisuMax 500. Additionally, this improved astigmatism correction may also be partially explained by the faster and more precise lenticule separation achieved with the VisuMax 800, which may reduce intraoperative variability and enhance treatment accuracy.

Current femtosecond laser platforms, including the VisuMax 800, do not yet incorporate active feedback systems for real-time aberration correction. Recent experimental studies have explored technologies such as adaptive optics and interferometry to improve optical precision. For example, Salter et al. demonstrated that adaptive optics can correct wavefront aberrations during laser fabrication [25], while Khorin et al. introduced a neural network-assisted interferometric method for more sensitive detection of asymmetric aberrations [26]. Although these approaches have not yet been applied in clinical platforms, they suggest potential directions for future system development.

This study has several limitations that warrant discussion. First, the overall number of participants was relatively small, and notably, the group treated with VisuMax 800 had fewer eyes compared to the VisuMax 500 group. This discrepancy is likely due to the recent introduction of SMILE surgery using VisuMax 800 in Korea, along with the lower price accessibility of VisuMax 800 compared to VisuMax 500. Despite this, it is significant that this study compared clinical outcomes post-SMILE surgery in consecutive patients treated at the same institution by a single surgeon during the same period. Secondly, the VisuMax 800 group tended to include patients with higher preoperative astigmatism levels. This may be attributed to the advanced features of the VisuMax 800, such as centration and cyclotorsion adjustment software, which likely made it the preferred choice for patients with high astigmatism. Third, this study compared two devices from the same manufacturer, which may introduce bias by unintentionally emphasizing the strengths of a specific product line. As a result, the findings may have limited generalizability. The devices were selected solely based on their availability at our institution, and the study was not intended for promotional purposes. Nonetheless, future research should include devices from multiple manufacturers, ideally through multi-center collaborations, to ensure more balanced and widely applicable evaluations. Fourth, we did not perform direct measurements of laser fluence at the corneal stromal plane. As this was a clinical comparative study focusing on surgical outcomes of the VisuMax 500 and 800 platforms, such benchtop optical measurements were beyond the scope of the protocol. Instead, we estimated the per-pulse fluence from the known pulse energy (*E_p_*) and the programmed spot and track distances (Δ*x* and Δ*y*) according to the equation *F* = *E_p_*/(Δ*x*·Δ*y*) [19,27]. As a result, the calculated fluence values for the VisuMax 500 and 800 were similar. A previous study has indicated that a simple conclusion stating that smaller spot and track distances combined with higher laser energy result in minimal roughness cannot be drawn, as multiple interacting factors influence the resultant surface quality [27]. Therefore, based on the data obtained in this study, it remains unclear whether the differences in clinical outcomes are attributable primarily to laser–tissue interaction characteristics or to improved device features such as centration guidance and cyclotorsion compensation. Future investigations integrating clinical outcomes with direct fluence measurements in optical laboratory settings may further elucidate the relationship between laser parameters and clinical results. Finally, as this study represents an initial report on the clinical outcomes of SMILE surgeries with VisuMax 800 in Korea, a notable limitation is the short follow-up duration of two months. Although previous studies have reported long-term stability of refractive outcomes following SMILE surgery [8,9,10,11], further clinical research involving a larger cohort and longer follow-up is necessary to compare the refractive correction efficacy of myopia and compound myopic astigmatism between VisuMax 500 and VisuMax 800.

## 5. Conclusions

The results of this study demonstrate that SMILE surgery using both VisuMax 500 and VisuMax 800, with existing nomograms, effectively corrected myopia. Post-surgery, 99% of eyes treated with VisuMax 500 and 97% of eyes treated with VisuMax 800 achieved a UDVA of 20/20. However, while VisuMax 500 was effective in correcting compound myopic astigmatism, VisuMax 800 showed superior precision in correcting both the magnitude and axis of astigmatism. Therefore, the ability of surgeons to achieve excellent results using existing nomograms with VisuMax 800 allows for an easy transition to adopting this technology for SMILE surgery. Particularly for patients with high astigmatism, SMILE with VisuMax 800 is expected to provide improved astigmatism correction outcomes, enhancing the results of corneal refractive surgery.

## Figures and Tables

**Figure 1 diagnostics-15-02450-f001:**
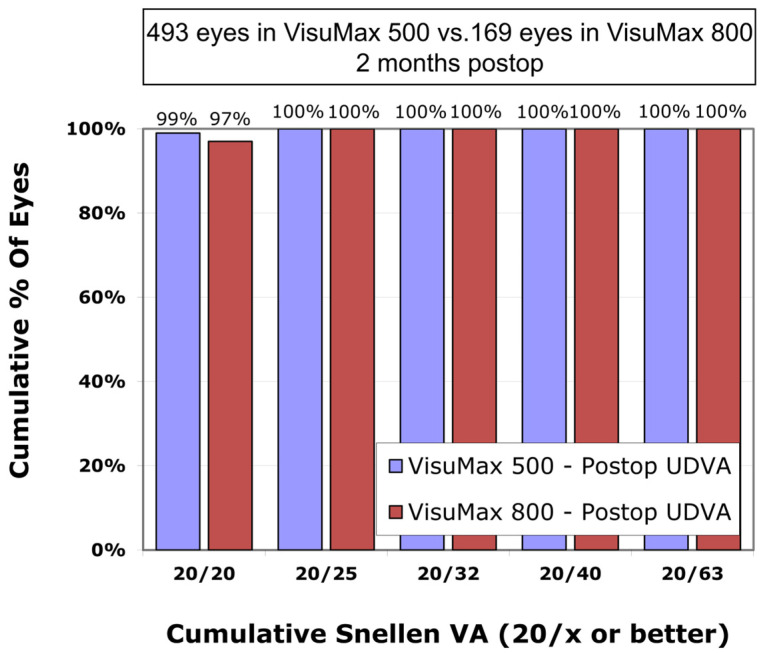
Comparison of postoperative cumulative Snellen visual acuity between the VisuMax 500 and 800 groups. UDVA, uncorrected distance visual acuity.

**Figure 2 diagnostics-15-02450-f002:**
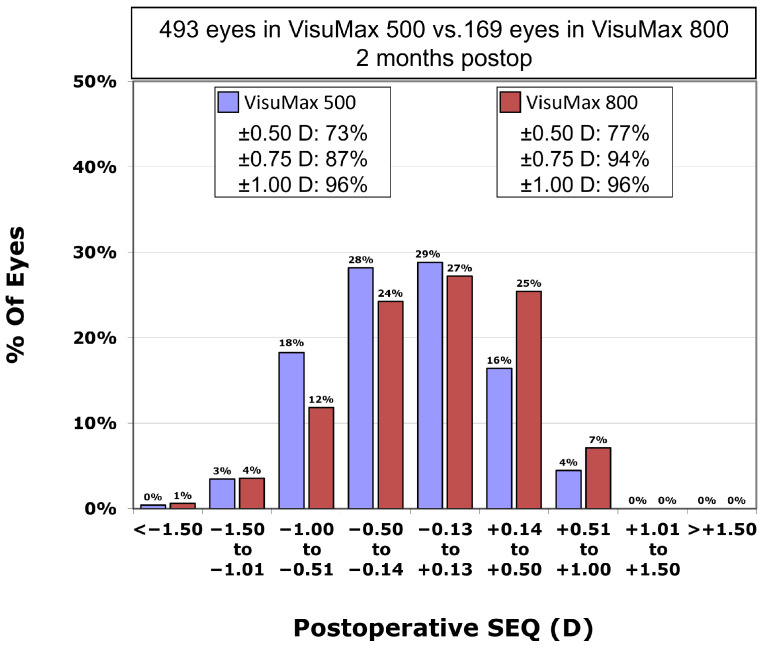
Comparison of postoperative spherical equivalent (SEQ) between the VisuMax 500 and 800 groups.

**Figure 3 diagnostics-15-02450-f003:**
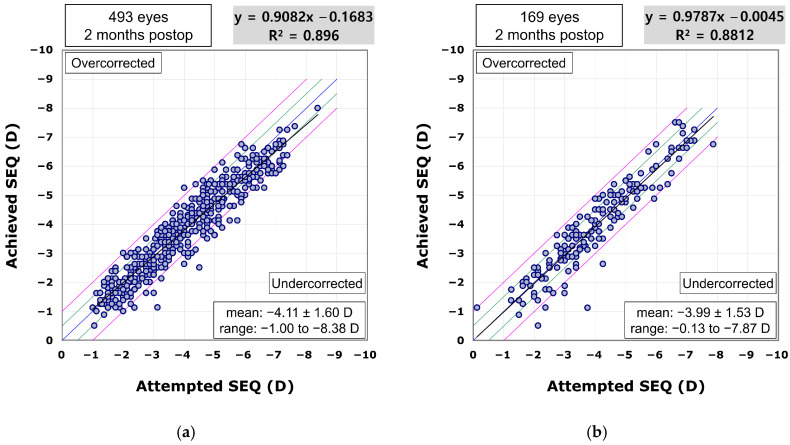
The achieved spherical equivalent (SEQ) plotted against the attempted SEQ in the VisuMax 500 (**a**) and 800 (**b**) groups.

**Figure 4 diagnostics-15-02450-f004:**
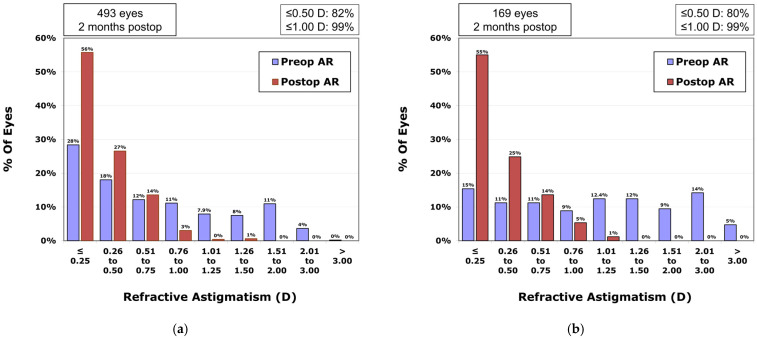
Comparison of preoperative and postoperative refractive astigmatism measured using an autorefractor in the VisuMax 500 (**a**) and 800 (**b**) groups.

**Figure 5 diagnostics-15-02450-f005:**
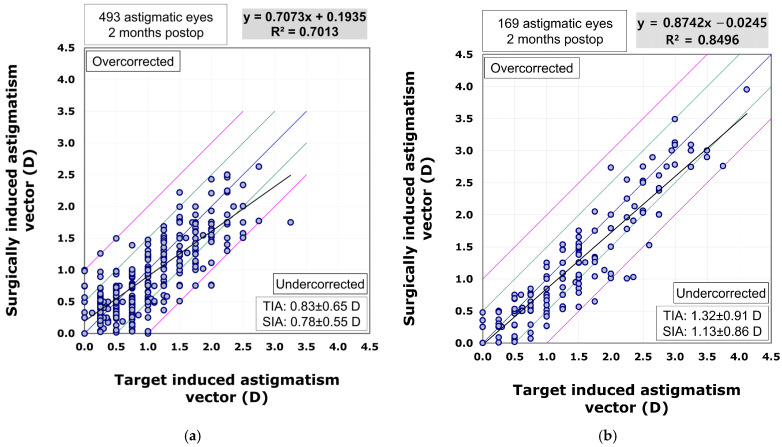
The surgically induced astigmatism vector plotted against the target induced astigmatism vector in the VisuMax 500 (**a**) and 800 (**b**) groups.

**Figure 6 diagnostics-15-02450-f006:**
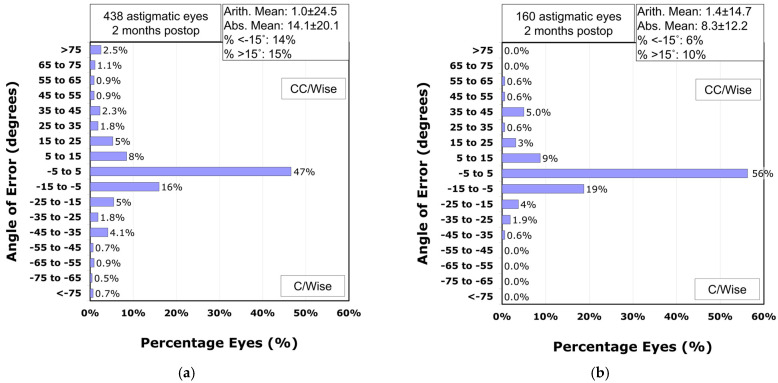
The refractive astigmatism angle of error in the VisuMax 500 (**a**) and 800 (**b**) groups.

**Table 1 diagnostics-15-02450-t001:** Nomogram adjustment for the amount of spherical manifest refraction to be corrected in SMILE surgery.

Measured Spherical Manifest Refraction (D)	Adjustment of Spherical Manifest Refraction (D)
0.00 to −1.00	Add −0.85
−1.00 to −2.00	Add −0.90
−2.00 to −3.00	Add −0.95
−3.00 to −4.00	Add −1.00
−4.00 to −5.00	Add −1.05
−5.00 to −6.00	Add −1.10
−6.00 to −7.00	Add −1.15
−7.00 to −8.00	Add −1.20 to −1.25
−8.00 to −9.00	Add −1.25 to −1.30

Patients aged 18–20 years received an additional −0.1 D of overcorrection to the spherical manifest refraction. No additional correction was made for patients aged 21–34 years. Patients between the ages of 35–40 years received an additional +0.1 D of undercorrection, and those aged 41 and above were adjusted with +0.2 D of undercorrection.

**Table 2 diagnostics-15-02450-t002:** Characteristics of Patients and Their Eyes in a Study Comparing VisuMax 500 and 800.

Parameters	VisuMax 500 (493 Eyes from 249 Patients)	VisuMax 800 (169 Eyes from 85 Patients)	*p*-Value *
Age, years	23.2 ± 4.6	24.0 ± 5.6	0.226
Sex, male:female, *n* (%)	91 (36.5):158 (63.5)	42 (49.4):43 (50.6)	0.041 ^†^
Laterality, right eye:left eye, *n* (%)	245 (49.7):248 (50.3)	84 (49.7):85 (50.3)	>0.999 ^†^
Preoperative visual acuity			
UDVA, logMAR	1.40 ± 0.68	1.31 ± 0.63	0.125
CDVA, logMAR	0.00 ± 0.00	0.00 ± 0.01	0.312
Manifest refractive error			
Sphere, D	−3.68 ± 1.51	−3.34 ± 1.54	0.012
Cylinder, CD	−0.94 ± 0.68	−1.30 ± 0.91	<0.001
Spherical equivalent, D	−4.09 ± 1.61	−4.00 ± 1.53	0.486
Pachymetry, μm	554.2 ± 28.6	552.7 ± 24.7	0.537
Optical zone, mm	6.4 ± 0.1	6.4 ± 0.1	0.921
Lenticule thickness, μm	100.2 ± 22.9	101.0 ± 21.3	0.708
Residual bed thickness, μm	344.1 ± 34.0	341.5 ± 31.2	0.384

UDVA, uncorrected distance visual acuity; CDVA, corrected distance visual acuity; D, diopters; CD, cylinder diopters. * Student’s *t*-test. ^†^ Fisher’s exact test.

## Data Availability

The data that support the findings of this study are available from the corresponding author upon reasonable request.

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
