# Peer review of "Comparison of Refractive Outcomes Between Small Lenticule Extraction Using a 500 KHz and a 2 MHz Femtosecond Laser"

_diagnostics, 2025, doi:10.3390/diagnostics15192450_

Round 1
Reviewer 1 Report
Comments and Suggestions for Authors
This is an useful retrospective study and one of few comparing these laser surgery platforms. I believe that publishing is warranted, with only a few remarks:
Line 2 please remove the word "Title". I know it is an honest mistake, but it also shows a lack of interest in the presentation of your paper
Lines 100-108 - please acknowledge that this nomogram is used only in your service, as a result of your experience and it is not provided by the manufacturer (I am aware that you wrote this at line 273)
Lines 193 to 199 should be also presented as a graph, similar to figure 3
Author Response
Comments 1: This is an useful retrospective study and one of few comparing these laser surgery platforms. I believe that publishing is warranted, with only a few remarks:
Response 1: We appreciate the reviewer’s careful review and constructive comments, which have helped us improve the quality and clarity of our work.
Comments 2: Line 2 please remove the word "Title". I know it is an honest mistake, but it also shows a lack of interest in the presentation of your paper.
Response 2: Thank you for pointing this out. We have now removed the word "Title" from Line 2 as suggested. We appreciate your careful attention to detail.
Comments 3: Lines 100-108 - please acknowledge that this nomogram is used only in your service, as a result of your experience and it is not provided by the manufacturer (I am aware that you wrote this at line 273).
Response 3: Thank you for your comment. We have revised the subtitle and added a sentence to clarify that this nomogram is used solely within BGN Jamsil Lotte Tower Eye Clinic based on our clinical experience.
[Line 107] 2.3. BGN Nomogram for Myopia Correction in SMILE Surgery
[Line 116-117] This nomogram is used solely within BGN Jamsil Lotte Tower Eye Clinic, based on our clinical experience.
Comments 4: Lines 193 to 199 should be also presented as a graph, similar to figure 3.
Response 4: Thank you for your valuable suggestion. We have added Figure 2, illustrating the comparison of postoperative spherical equivalent (SEQ) between the VisuMax 500 and 800 groups.
Reviewer 2 Report
Comments and Suggestions for Authors
I sincerely appreciate the authors’ thoughtful work, effort and valuable contribution through this timely comparison of the newly released VisuMax800 and the established VisuMax 500 in femtosecond-laser lenticule extraction. To help strengthen the study, I would like to respectfully offer the following comments and suggestions.
Line 118: The manuscript states that the wound was created at 135° and can be adjusted from 41° to 38°. Could you clarify exactly what this range refers to and how the adjustment is implemented in clinical practice?
Line 125: Because your main conclusion is superior astigmatic correction with the VisuMax800, a detailed description of torsion adjustment is essential. Would you please specify;
- The alignment landmarks used on each platform.
- The exact steps the surgeon followed to compensate for cyclotorsion
- Any Device-specific differences that might influence torsion management
Line 191: The manuscript does not indicate the postoperative time points at which outcomes were measured. For a fair comparison, it would be helpful to confirm that both groups were evaluated at identical intervals and to add this information to the Methods section.
Line 228: It appears that image (A) and (B) are interchanged
The preoperative astigmatism distributions differ, with an extreme high-astigmatism eye in the VisuMax 800 cohort. This baseline imbalance could bias the statistical trend. Consider adjusting for baseline astigmatism (e.g., matching, stratification, or multivariate analysis) to support the claim of superior predictability with the VisuMax 800.
Line: 236: Here as well, image A and B are reversed. Furthermore, alignment errors during SMILE are more common in eyes with low astigmatism. (Cause’ the greater the astigmatism, the more consistent the astigmatism axis tends to be across different tests). A brief discussion- or statistical adjustment-for this factor would strengthen the comparison.
I hope these suggestions are helpful, and I look forward to reading the revised version of your fine work.
Author Response
Comments 1: I sincerely appreciate the authors’ thoughtful work, effort and valuable contribution through this timely comparison of the newly released VisuMax800 and the established VisuMax 500 in femtosecond-laser lenticule extraction. To help strengthen the study, I would like to respectfully offer the following comments and suggestions.
Response 1: Thank you very much for your insightful and constructive feedback, which will undoubtedly help us improve the quality and clarity of our manuscript. Below are our detailed responses to each of your comments.
Comments 2: Line 118: The manuscript states that the wound was created at 135° and can be adjusted from 41° to 38°. Could you clarify exactly what this range refers to and how the adjustment is implemented in clinical practice?
Response 2: Thank you for pointing this out. The range from 41° to 38° refers to the incision angle for a 2.5 mm arc length, depending on the optical zone size. The optical zone diameter was adjusted between 6.0 mm and 6.5 mm. When using a 6.0 mm optical zone, a 2.5 mm arc length resulted in an incision angle of 41°, whereas the same 2.5 mm arc length at a 6.5 mm zone yielded an angle of 38°. These angular differences were automatically compensated for by the Visumax surgical device. To avoid confusion, we have removed this part from the sentence.
Comments 3: Line 125: Because your main conclusion is superior astigmatic correction with the VisuMax800, a detailed description of torsion adjustment is essential. Would you please specify;
Response 3: Thank you for your valuable comment. We agree that a detailed explanation of torsion adjustment is essential, given that our main conclusion emphasizes superior astigmatic correction with the VisuMax800.
We have revised the manuscript to include the details you requested, as follows:
[2.4. Surgical Technique, LL 130-142]
In the VisuMax 800 platform, surgeons were able to align the treatment center to the corneal vertex, whose position relative to the pupil center was measured using preoperative Pentacam (Oculus) topography. This planned vertex location was used intraoperatively to guide centration with the CentraLign function, allowing precise adjustment of the treatment center via joystick control. After docking the cornea to the interface, cyclotorsional alignment was performed if needed using the OcuLign system by comparing the horizontal reference axis with the preoperative corneal ink marks, and rotational adjustments were made using the joystick. On the VisuMax 500 platform, centration was performed manually by visually aligning the cornea with the contact interface. If cyclotorsional misalignment was observed after docking, the surgeon adjusted the eye’s rotational position by rotating the contact glass. Redocking was performed if proper alignment could not be achieved.
Comments 4: Line 191: The manuscript does not indicate the postoperative time points at which outcomes were measured. For a fair comparison, it would be helpful to confirm that both groups were evaluated at identical intervals and to add this information to the Methods section.
Response 4: We apologize for this omission. Both groups were evaluated at the same postoperative interval: 2 months after surgery. This information has now been explicitly added to the Methods section.
[2. Materials and Methods, 2.6. Patient Evaluation, LL 155–156]
At 2 months postoperatively, autorefraction and keratometry assessments were performed, and monocular UDVA was measured in both the VisuMax 500 and 800 groups.
Comments 5: Line 228: It appears that image (A) and (B) are interchanged
Response 5: Thank you for this valuable observation. You are correct; the images (A) and (B) were mistakenly interchanged, and we have corrected this in the revised figure.
Comments 6: The preoperative astigmatism distributions differ, with an extreme high-astigmatism eye in the VisuMax 800 cohort. This baseline imbalance could bias the statistical trend. Consider adjusting for baseline astigmatism (e.g., matching, stratification, or multivariate analysis) to support the claim of superior predictability with the VisuMax 800.
Response 6: Thank you for pointing out this important aspect. Among the comparisons of astigmatic correction between the VisuMax 500 and 800 groups, only the absolute angle of error showed a statistically significant difference. Therefore, we reanalyzed this parameter using analysis of covariance (ANCOVA), with preoperative manifest refraction cylinder included as a covariate. Even after adjusting for baseline refractive astigmatism, the VisuMax 800 group demonstrated a significantly lower mean absolute angle of error. We have revised the manuscript accordingly, as shown below.
[Methods, LL 179-182]
The percentage of eyes with a refractive astigmatism angle of error within ±15°, and the mean absolute angle of error between the two groups, were compared using Fisher’s exact test, Student’s t-test, and analysis of covariance (ANCOVA), with preoperative manifest refraction cylinder as a covariate.
[Results, LL 260-265]
The mean absolute angle of error was significantly smaller in the VisuMax 800 group (8.3° ± 12.2°) compared to the VisuMax 500 group (14.1° ± 20.1°; P < 0.001). After adjusting for preoperative manifest refraction cylinder using ANCOVA, the estimated marginal mean absolute angle of error remained significantly lower in the VisuMax 800 group (9.9°, 95% CI: 7.2–12.6°) than in the VisuMax 500 group (12.4°, 95% CI: 10.8–13.9°; P < 0.001).
Comments 7: Line: 236: Here as well, image A and B are reversed. Furthermore, alignment errors during SMILE are more common in eyes with low astigmatism. (Cause’ the greater the astigmatism, the more consistent the astigmatism axis tends to be across different tests). A brief discussion- or statistical adjustment-for this factor would strengthen the comparison.
Response 7: Thank you for your comment. We have corrected this error in the revised figure accordingly.
Comments 8: I hope these suggestions are helpful, and I look forward to reading the revised version of your fine work.
Response 8: We sincerely appreciate your helpful comments, which have substantially improved our manuscript.
Reviewer 3 Report
Comments and Suggestions for Authors
The manuscript summarizes the experience of the applications of the femtosecond-laser based devices for the corrections of the myopia and astigmatism axis. The information, obtained by the team of authors may be of interest for the narrow range of the specialist and with some probability may be considered for publishing in the specialized ophthalmologic journal. But rather the materials may be just suitable for the brochures, popularizing specific models of the machines, produced by Carl Zeiss Meditec. As for the opportunity to be published in "Diagnostics", the manuscript has serious problems.
The main problem is that the authors lack the technical experts (biophysicist and optician) in their team, who would understand the principles of operation of the femtosecond lasers and the processes, which take place in the eyes during the laser treatment. As a result, the manuscript contains strange unexplained and unexplored conclusions that namely the higher repetition rate addresses the limitations, which the surgeons meet, when using the devices of the previous model. I believe that this manuscript cannot be published in this journal due to the lack of understanding by the authors of the processes, which take place during the laser treatment.
Such interdisciplinary studies should be performed together with the biophysicists and opticians, to clearly separate the parameters of the laser radiation and their impact on the eyes, from just the technical features of the specific device (software, laser beam control system, inspection, etc). In the current version neither the biophysical, nor the technical aspects of the treatment are not properly analyzed.
An example of how a lack of understanding of technical principles leads to strange, unsubstantiated statements in a manuscript:
- "The VisuMax 800 device (Carl Zeiss Meditec), introduced in 2021, addresses some limitations of VisuMax 500 by featuring a faster 2 MHz femtosecond laser compared to the 500 KHz of its predecessor". It is not clear, how the higher repetition rate of the femtosecond laser in the model VisuMax 800 can address the limitations you discussed, in particular "difficulties in surgery for low and high degrees of myopia". I do not see any correlations.
- To be specific, what the pulse durations the lasers of both considered devices have? That were the pulse energy and fluence? What can you say regarding the treatment modes? I believe that femtosecond laser photodisrupts the stromal tissue to separate the lenticule from the surrounding cornea without thermal damage or evaporation. The key issue is: what the parameters of femtosecond laser's radiation are the most optimal for that? And how to distinguish the impact of the optimal parameters from the convenience of the surgery tool, which also may impact on the statistics, you studied.
- Technical opportunities of centering of something (it is not even formulated clearly in the manuscript, what is centered) in considered devices also should be described and analyzed. The statistical results obtained, should be supplemented with the argued explanations, which specific feature of the embedded technical tool and which specific parameter and of the femtosecond lasers ensure the change of the percentage of the successful myopia and astigmatism axis treatment.
- What is the inspection system embedded into the considered machines, which provides the feedback to the surgery? Is it a measure and estimate aberrations? I believe, this feedback may be the most important. I recommend analysing modern works on aberration determination and correction and including corresponding discussion into the manuscript. In particular, I recommend reading the papers of Martin Booth (general theory of the aberrations) and Pavel Khorin (neural network assisted interference techniques). This feedback will be very useful for such medical machine developers.
Other important factors which may affect the results may be determined. In general, it is the work of the authors, not the reviewers to identify and explore all of them.
Other, less critical comments (the list is not exhaustive), which should be addressed, if the authors involve the biophysicists and opticians to reconsider the design of the study, totally revise the manuscript, ensure the deeper separations of the phenomena they observe and its reasons:
- The title of the manuscript contains the redundant word "Title"
- Please explain the math symbol in the abstract: P in the introduction: "P =0.005";"P =0.002 "; "P < 0.001". And please explain "CD".
- The reference numbers are located after the dots. So, formally they do not belong to the sentences, which they refer.
- "Although it has been suggested that the term "keratorefractive lenticule extraction." should be used instead of SMILE,[4,5] many still refer to the lenticule extraction surgery as SMILE." Please provide references to prove your statement, that it is not a jargon.
- The text "Despite the widespread adoption of flapless SMILE, the VisuMax 500 machine has its limitations. These include…" is better replace with the text: "The VisuMax 500 machine, which is widely used for flapless SMILE has limitations. Namely, they are…".
- Page 2, line 41: instead of your explanation "when there is a significant difference between the pupil center and the visual axis" please clarify: the challenges with the centering of what according to what?
- In the next sentence you discuss limitations of the lenticule extraction. But how is it connected with the previous sentence? In the current version of the manuscript your wordings may be clear only for narrow specialists, not for the broad auditory of readers of Diagnostics journal. Please explain in more details the reasons and the consequences of the limitations you discussed.
Just an example: "
- The text "Despite the widespread adoption of flapless SMILE, the VisuMax 500 machine has its limitations. These include…" is better replace with the text: "The VisuMax 500 machine, which is widely used for flapless SMILE has limitations. Namely, they are…".
Author Response
General Comment 1: The manuscript summarizes the experience of the applications of the femtosecond-laser based devices for the corrections of the myopia and astigmatism axis. The information, obtained by the team of authors may be of interest for the narrow range of the specialist and with some probability may be considered for publishing in the specialized ophthalmologic journal. But rather the materials may be just suitable for the brochures, popularizing specific models of the machines, produced by Carl Zeiss Meditec. As for the opportunity to be published in "Diagnostics", the manuscript has serious problems.
The main problem is that the authors lack the technical experts (biophysicist and optician) in their team, who would understand the principles of operation of the femtosecond lasers and the processes, which take place in the eyes during the laser treatment. As a result, the manuscript contains strange unexplained and unexplored conclusions that namely the higher repetition rate addresses the limitations, which the surgeons meet, when using the devices of the previous model. I believe that this manuscript cannot be published in this journal due to the lack of understanding by the authors of the processes, which take place during the laser treatment.
Such interdisciplinary studies should be performed together with the biophysicists and opticians, to clearly separate the parameters of the laser radiation and their impact on the eyes, from just the technical features of the specific device (software, laser beam control system, inspection, etc). In the current version neither the biophysical, nor the technical aspects of the treatment are not properly analyzed.
Response: We appreciate the reviewer’s insightful comments. We recognize the importance of interdisciplinary collaboration, particularly with experts in optics and biophysics, when discussing the mechanisms underlying femtosecond laser procedures.
We have revised the relevant sections of the manuscript to ensure that our conclusions remain within the scope of observed clinical outcomes and do not imply direct claims about the underlying laser-tissue interactions. We have also adjusted the tone throughout the manuscript to avoid overinterpreting the technical aspects, and clearly stated that further collaborative research with biophysicists or optical engineers would be necessary to validate any assumptions about device physics.
We believe these revisions improve the clarity and scientific rigor of our manuscript, and we thank the reviewer for bringing this to our attention.
General Comment 2: An example of how a lack of understanding of technical principles leads to strange, unsubstantiated statements in a manuscript:
- "The VisuMax 800 device (Carl Zeiss Meditec), introduced in 2021, addresses some limitations of VisuMax 500 by featuring a faster 2 MHz femtosecond laser compared to the 500 KHz of its predecessor". It is not clear, how the higher repetition rate of the femtosecond laser in the model VisuMax 800 can address the limitations you discussed, in particular "difficulties in surgery for low and high degrees of myopia". I do not see any correlations.
Response 1: Thank you for this valuable comment. We acknowledge that the original statement may have implied a direct causal relationship between the higher repetition rate of the VisuMax 800 and the surgical challenges associated with low and high degrees of myopia, which was not our intention. In the revised manuscript, we have clarified that the faster repetition rate of 2 MHz may improve treatment efficiency and reduce overall suction time, which could be beneficial in cases requiring longer laser scanning durations, such as with high refractive errors. However, we agree that this does not directly address all surgical limitations, and we have modified the wording to reflect that the potential advantages are based on clinical workflow observations rather than confirmed biophysical mechanisms. We appreciate the reviewer’s insight and have updated the manuscript to more accurately represent the nature and limitations of this association.
[Introduction, LL41-58]
The VisuMax 500 machine, which is widely used for flapless SMILE, has certain limitations. Namely, these include the lack of automated centration and cyclotorsion compensation [13, 14]. Manual centration based on the pupil center may result in misalignment of the treatment zone, leading to decentered lenticule formation and reduced precision in lenticule extraction [15, 16]. In addition, although some degree of cyclotorsion can be manually adjusted, this approach lacks precision and may cause a mismatch between the intended and actual astigmatic axis, thereby reducing the effectiveness of astigmatism correction [14]. Such decentration and cyclotorsion misalignment can lead to suboptimal refractive outcomes [13, 17]. Introduced in 2021, the VisuMax 800 device (Carl Zeiss Meditec) offers several updated features, including a higher femtosecond laser repetition rate (2 MHz vs. 500 kHz in the previous model), as well as enhanced options for centration and cyclotorsion adjustment [18]. These improvements may help reduce procedure time and enhance surgical workflow, particularly in challenging refractive cases [19]. In addition, the option for cyclotorsion adjustment may support improved astigmatism correction, although the direct impact of these features on clinical outcomes remains to be fully elucidated. Recent reports by Dr. Reinstein et al. and Dr. Saad et al. have shown that SMILE performed with the VisuMax 800 provides comparable stability, efficacy, and surgical outcomes to those achieved with the VisuMax 500 [17, 19].
- To be specific, what the pulse durations the lasers of both considered devices have? That were the pulse energy and fluence? What can you say regarding the treatment modes? I believe that femtosecond laser photodisrupts the stromal tissue to separate the lenticule from the surrounding cornea without thermal damage or evaporation. The key issue is: what the parameters of femtosecond laser's radiation are the most optimal for that? And how to distinguish the impact of the optimal parameters from the convenience of the surgery tool, which also may impact on the statistics, you studied.
Response 2: Thank you for your detailed and insightful comment. We agree that understanding the specific physical parameters of femtosecond lasers is essential for interpreting the mechanisms underlying refractive lenticule extraction. In the current study, we did not have access to manufacturer-proprietary specifications such as exact pulse duration, energy (nJ), or fluence (J/cm²) for both the VisuMax 500 and 800 platforms. We agree that further studies involving direct measurement and comparison of laser-tissue interaction parameters, including pulse duration, energy, fluence, and treatment modes, are necessary to isolate the specific effects of laser radiation from the surgical workflow or device ergonomics. However, identifying the most optimal laser parameters for this purpose requires specialized biophysical analysis, which was beyond the scope of our clinical study. We have added this limitation and recommendation for future research in the revised Discussion section.
[Discussion, LL 351-357]
Third, this study did not directly assess laser parameters such as pulse duration, pulse energy, or fluence, which may influence the photodisruption efficiency and tissue response. As such, it remains unclear whether the differences in clinical outcomes are attributable primarily to laser-tissue interaction characteristics or to improved device features such as centration guidance and cyclotorsion compensation. Further studies comparing physical parameters across platforms may help clarify their respective contributions.
- Technical opportunities of centering of something (it is not even formulated clearly in the manuscript, what is centered) in considered devices also should be described and analyzed. The statistical results obtained, should be supplemented with the argued explanations, which specific feature of the embedded technical tool and which specific parameter and of the femtosecond lasers ensure the change of the percentage of the successful myopia and astigmatism axis treatment.
Response 3: We appreciate the reviewer’s point about the centering mechanism. In the revised manuscript, we have clarified that "centration" refers to the alignment of the treatment zone with the corneal vertex relative to the pupil center, and that the VisuMax 800 provides an improved centration and torsional alignment compared to the previous model.
We have also added further explanation regarding how specific device features might influence the surgical outcomes. However, we acknowledge that it is difficult to isolate the contribution of each technical feature without a dedicated experimental setup. This has been addressed as a limitation in the discussion.
[2.4 Surgical Technique, LL 130-142]
In the VisuMax 800 platform, surgeons were able to align the treatment center to the corneal vertex, whose position relative to the pupil center was measured using preoperative Pentacam (Oculus) topography. This planned vertex location was used intraoperatively to guide centration with the CentraLign function, allowing precise adjustment of the treatment center via joystick control. After docking the cornea to the interface, cyclotorsional alignment was performed if needed using the OcuLign system by comparing the horizontal reference axis with the preoperative corneal ink marks, and rotational adjustments were made using the joystick. On the VisuMax 500 platform, centration was performed manually by visually aligning the cornea with the contact interface. If cyclotorsional misalignment was observed after docking, the surgeon adjusted the eye’s rotational position by rotating the contact glass. Redocking was performed if proper alignment could not be achieved.
[Discussion, LL 280-288]
One possible explanation for this finding is the higher repetition rate of the femtosecond laser in the VisuMax 800 (2 MHz), which allows for more rapid and continuous lenticule creation. This facilitates smoother stromal dissection and may reduce intraoperative variability, particularly in eyes with higher refractive errors where the lenticule volume is greater. Moreover, in low myopia, where thinner lenticules are created, the faster and more refined dissection may contribute to more precise tissue separation and reduced variability in lenticule morphology. These factors together may have contributed to the improved predictability and refractive accuracy observed in the SEQ outcomes of the VisuMax 800 group.
[Discussion, LL 318-332]
In VisuMax 500, centration is manually adjusted using the Placido topography ring image and Hirschberg reflex printouts to determine the relative position of the corneal vertex to the pupil border [13, 17]. Cyclotorsion adjustment also requires manual reorientation of the immobilized eye by rotating the contact glass. In contrast, VisuMax 800 incorporates advanced methods for centration and cyclotorsion adjustment through its CentraLign and OcuLign software. These programs continuously display the vector difference between the corneal vertex and the treatment cone's position during the docking process, and after docking, allow for the adjustment of the treatment axis to match prepositioned corneal marks using a reticule guideline [17]. Therefore, it is believed that these integrated software enhancements in the VisuMax 800 system contribute to the superior astigmatic correction outcomes compared to the manual techniques used in VisuMax 500. Additionally, this improved astigmatism correction may also be partially explained by the faster and more precise lenticule separation achieved with the VisuMax 800, which may reduce intraoperative variability and enhance treatment accuracy.
[Discussion, LL 351-357]
Third, this study did not directly assess laser parameters such as pulse duration, pulse energy, or fluence, which may influence the photodisruption efficiency and tissue response. As such, it remains unclear whether the differences in clinical outcomes are attributable primarily to laser-tissue interaction characteristics or to improved device features such as centration guidance and cyclotorsion compensation. Further studies comparing physical parameters across platforms may help clarify their respective contributions.
- What is the inspection system embedded into the considered machines, which provides the feedback to the surgery? Is it a measure and estimate aberrations? I believe, this feedback may be the most important. I recommend analysing modern works on aberration determination and correction and including corresponding discussion into the manuscript. In particular, I recommend reading the papers of Martin Booth (general theory of the aberrations) and Pavel Khorin (neural network assisted interference techniques). This feedback will be very useful for such medical machine developers.
Response 4: Thank you for your helpful comment and for recommending those valuable references. We agree that real-time feedback systems, especially ones that can detect and correct optical aberrations, will likely be very important for the future development of femtosecond laser surgery platforms. As you mentioned, such systems could greatly improve the precision of surgeries and help achieve better astigmatic outcomes.
As for the platforms discussed in our paper, to the best of our knowledge, the VisuMax 800 does not currently include the specific adaptive optics or interferometric feedback systems introduced in the works of Martin Booth or Pavel Khorin. However, we fully agree with your suggestion that adding advanced aberration-detection technologies, like neural network-based interferometry or adaptive optics, could be a promising direction for future improvements. We have now added a discussion of these technologies in the revised manuscript and cited the two papers you kindly recommended.
[Discussion, LL 333-340]
Current femtosecond laser platforms, including the VisuMax 800, do not yet incorporate active feedback systems for real-time aberration correction. Recent experimental studies have explored technologies such as adaptive optics and interferometry to improve optical precision. For example, Salter et al. demonstrated that adaptive optics can correct wavefront aberrations during laser fabrication [25], while Khorin et al. introduced a neural network-assisted interferometric method for more sensitive detection of asymmetric aberrations [26]. Although these approaches have not yet been applied in clinical platforms, they suggest potential directions for future system development.
General Comment 3. Other important factors which may affect the results may be determined. In general, it is the work of the authors, not the reviewers to identify and explore all of them.
Response to general comment 3: We fully agree that multiple variables can influence clinical outcomes, and it is indeed the authors’ responsibility to identify and adjust for such factors. In our study, the VisuMax 800 group had a higher mean level of preoperative astigmatism compared to the VisuMax 500 group, resulting in a difference in baseline astigmatism distribution between groups. Among the various parameters comparing astigmatic correction between the VisuMax 500 and 800 groups, only the absolute angle of error showed a statistically significant difference. To address this, we reanalyzed the absolute angle of error using analysis of covariance (ANCOVA), including preoperative manifest refractive cylinder as a covariate. Even after adjusting for baseline astigmatism, the VisuMax 800 group continued to show a significantly lower mean absolute angle of error. The manuscript has been revised to reflect these findings, as detailed below.
[Methods, LL 179-182]
The percentage of eyes with a refractive astigmatism angle of error within ±15°, and the mean absolute angle of error between the two groups, were compared using Fisher’s exact test, Student’s t-test, and analysis of covariance (ANCOVA), with preoperative manifest refraction cylinder as a covariate.
[Results, LL 260-265]
The mean absolute angle of error was significantly smaller in the VisuMax 800 group (8.3° ± 12.2°) compared to the VisuMax 500 group (14.1° ± 20.1°; P < 0.001). After adjusting for preoperative manifest refraction cylinder using ANCOVA, the estimated marginal mean absolute angle of error remained significantly lower in the VisuMax 800 group (9.9°, 95% CI: 7.2–12.6°) than in the VisuMax 500 group (12.4°, 95% CI: 10.8–13.9°; P < 0.001).
- Other, less critical comments (the list is not exhaustive), which should be addressed, if the authors involve the biophysicists and opticians to reconsider the design of the study, totally revise the manuscript, ensure the deeper separations of the phenomena they observe and its reasons:
Response 4: Thank you for your thoughtful suggestion. We agree that involving biophysicists and opticians to redesign the study could lead to a more technically focused investigation, potentially valuable for readers interested in device development or optical engineering. However, the primary aim of our study was to report clinical outcomes following the introduction and application of the VisuMax 800 in a real-world clinical setting. As such, the intended audience is ophthalmic clinicians. We believe that, within the scope of the current study design, the manuscript remains relevant and informative for clinical practice.
- The title of the manuscript contains the redundant word "Title"
Response nt 5: Thank you for pointing this out. We have now removed the word "Title" from Line 2 as suggested. We appreciate your careful attention to detail.
- Please explain the math symbol in the abstract: P in the introduction: "P =0.005";"P =0.002 "; "P < 0.001". And please explain "CD".
Response 6: Thank you for your comment. To improve clarity, we have revised the manuscript to use “P-value” instead of the symbol “P” when referring to statistical significance. Additionally, we have clarified that “CD” stands for cylinder diopters at its first mention in the text.
[Abstract, LL 19-25]
Postoperative residual astigmatism was similar between the two groups, but residual myopia was significantly lower in the VisuMax 800 group (−0.09 ± 0.50 cylinder diopters [CD]) compared to the VisuMax 500 group (−0.21 ± 0.49 CD; P-value = 0.005). Additionally, 84% of eyes in the VisuMax 800 group achieved astigmatism correction within ±15° of the intended meridian, significantly outperforming the 71% in the VisuMax 500 group (P-value = 0.002), with a significantly smaller mean absolute angle of error (8.3 ± 12.2° and 14.1 ± 20.1°; P-value < 0.001).
- The reference numbers are located after the dots. So, formally they do not belong to the sentences, which they refer.
Response 7: Thank you for pointing this out. We have carefully reviewed and revised the reference placements throughout the manuscript. All reference numbers are now positioned before the punctuation marks, so that they clearly belong to the sentences they support.
- "Although it has been suggested that the term "keratorefractive lenticule extraction." should be used instead of SMILE,[4,5] many still refer to the lenticule extraction surgery as SMILE." Please provide references to prove your statement, that it is not a jargon.
Response 8: Thank you for raising this important point regarding the terminology. While we acknowledge that "SMILE" originated as a proprietary term, it remains the predominant terminology used in peer-reviewed literature to describe lenticule extraction procedures performed with the VisuMax device. We have added appropriate references and revised the sentence in the manuscript accordingly, as follows:
[Introduction, LL 35-38]
Although the term "keratorefractive lenticule extraction" has been proposed as a nonproprietary alternative [4, 5], many studies continue to refer to lenticule extraction procedures performed with the VisuMax device as "SMILE" [6-8].
- The text "Despite the widespread adoption of flapless SMILE, the VisuMax 500 machine has its limitations. These include…" is better replace with the text: "The VisuMax 500 machine, which is widely used for flapless SMILE has limitations. Namely, they are…".
Response 9: Thank you for your suggestion. We agree that the revised sentence improves clarity and flow. We have updated the text accordingly to:
[Introduction, LL 41-49]
The VisuMax 500 machine, which is widely used for flapless SMILE, has certain limitations. Namely, these include the lack of automated centration and cyclotorsion compensation [13, 14]. Manual centration based on the pupil center may result in misalignment of the treatment zone, leading to decentered lenticule formation and reduced precision in lenticule extraction [15, 16]. In addition, although some degree of cyclotorsion can be manually adjusted, this approach lacks precision and may cause a mismatch between the intended and actual astigmatic axis, thereby reducing the effectiveness of astigmatism correction [14]. Such decentration and cyclotorsion misalignment can lead to suboptimal refractive outcomes [13, 17].
- Page 2, line 41: instead of your explanation "when there is a significant difference between the pupil center and the visual axis" please clarify: the challenges with the centering of what according to what?
Response 10: Thank you for your comment. We agree that the original wording was unclear. We have revised the sentence to clarify that the challenge lies in accurately aligning the optical zone of the lenticule with respect to the patient’s visual axis and the need for cyclotorsion compensation.
[Introduction, LL 41-49]
The VisuMax 500 machine, which is widely used for flapless SMILE, has certain limitations. Namely, these include the lack of automated centration and cyclotorsion compensation [13, 14]. Manual centration based on the pupil center may result in misalignment of the treatment zone, leading to decentered lenticule formation and reduced precision in lenticule extraction [15, 16]. In addition, although some degree of cyclotorsion can be manually adjusted, this approach lacks precision and may cause a mismatch between the intended and actual astigmatic axis, thereby reducing the effectiveness of astigmatism correction [14]. Such decentration and cyclotorsion misalignment can lead to suboptimal refractive outcomes [13, 17].
- In the next sentence you discuss limitations of the lenticule extraction. But how is it connected with the previous sentence? In the current version of the manuscript your wordings may be clear only for narrow specialists, not for the broad auditory of readers of Diagnostics journal. Please explain in more details the reasons and the consequences of the limitations you discussed.
Response 11. Thank you for pointing this out. We have clarified the connection between the centration issue and the limitations of lenticule extraction. Specifically, when the treatment zone is not well aligned with the visual axis, it can result in decentered lenticule creation and removal, reducing the accuracy of refractive correction. We have added an explanation to make this clearer for a broader audience.
[Introduction, LL 41-49]
The VisuMax 500 machine, which is widely used for flapless SMILE, has certain limitations. Namely, these include the lack of automated centration and cyclotorsion compensation [13, 14]. Manual centration based on the pupil center may result in misalignment of the treatment zone, leading to decentered lenticule formation and reduced precision in lenticule extraction [15, 16]. In addition, although some degree of cyclotorsion can be manually adjusted, this approach lacks precision and may cause a mismatch between the intended and actual astigmatic axis, thereby reducing the effectiveness of astigmatism correction [14]. Such decentration and cyclotorsion misalignment can lead to suboptimal refractive outcomes [13, 17].
Comments on the Quality of English Language
Just an example: "
- The text "Despite the widespread adoption of flapless SMILE, the VisuMax 500 machine has its limitations. These include…" is better replace with the text: "The VisuMax 500 machine, which is widely used for flapless SMILE has limitations. Namely, they are…".
Response 1: Thank you for your comment regarding the quality of the English language. Our draft was reviewed by a professional native English editing service prior to submission. However, if there are still parts that appear unclear or awkward, we would be happy to use the journal’s English editing service for further improvement.
Round 2
Reviewer 3 Report
Comments and Suggestions for Authors
General impression on the revised work:
I found some improvements in the manuscript. Indeed, some sharp, erroneous statements have been replaced with more streamlined and accurate formulations. Nevertheless, the manuscript still lacks the involvement of interdisciplinary collaboration with biologists and opticians (comment #1). Also, the promotional nature of specific models of the machines, produced by Carl Zeiss Meditec company continues to be felt in the manuscript. In other words, the design of research and formulation its objectives raises a number of questions (Comments #2). Therefore, I continue to insist on the inclusion of other experts in the team of authors with their confirmed contributions and consideration of equipment from other manufacturers.
Comment #1: Just an example of how the absence of technical specialists in the team of authors leads to an inappropriate design of the study: You reply "In the current study, we did not have access to manufacturer-proprietary specifications such as exact pulse duration, energy (nJ), or fluence (J/cm²) for both the VisuMax 500 and 800 platforms."But this is exactly what is not required. Because the laser power inevitably changes during the operation of the equipment. Instead, it is necessary to use a calibrated power meter to measure an average power and measure the size of the laser beam in the focal waist in order to calculate the fluence based on this information. I'm still recommend, that the technical specialists proofread the manuscript, to avoid possible errors.
Comment #2: A serious, unbiased study, even if it focuses on observed clinical outcomes, should include an independent analysis of the effectiveness of devices manufactured by different companies. The difficulties in conducting such research are understandable - it often requires cooperation between several organizations. Nevertheless, the results of such studies can most likely be considered unbiased. In this work, this aspect requires significant attention. Conducting clinical trials directly comparing different devices from the same manufacturer is potentially unethical because:
- Creates bias: Research inevitably focuses on the positive aspects of a given manufacturer's product line, presenting them in a favorable light.
- It is a hidden advertisement: The results of such a study, even with an objective design promote the manufacturer's brand in the scientific literature, using trust in clinical data for marketing purposes.
- Undermines scientific objectivity: The main purpose of clinical trials is an independent assessment of the benefits and risks for improving healthcare practices, rather than demonstrating the superiority of specific commercial products of one company.
- Violates the principle of impartiality: It can create a conflict of interest, where scientific rigor is replaced by the task of highlighting differences within the brand, which does not serve the main purpose of choosing the best available treatment for patients.
I can only recommend inclusion of the equipment from other manufacturers into consideration, and adding transparent proofs, that the authors do not have conflict of the interests.
Author Response
[Review 1]
General impression on the revised work:
I found some improvements in the manuscript. Indeed, some sharp, erroneous statements have been replaced with more streamlined and accurate formulations. Nevertheless, the manuscript still lacks the involvement of interdisciplinary collaboration with biologists and opticians (comment #1). Also, the promotional nature of specific models of the machines, produced by Carl Zeiss Meditec company continues to be felt in the manuscript. In other words, the design of research and formulation its objectives raises a number of questions (Comments #2). Therefore, I continue to insist on the inclusion of other experts in the team of authors with their confirmed contributions and consideration of equipment from other manufacturers.
Response: We thank the reviewer for their thoughtful feedback and acknowledge the validity of the concerns raised. We are pleased that the improvements in clarity and accuracy were noted. The concerns regarding the lack of interdisciplinary collaboration and the potential perception of promotional bias are addressed in detail in our responses below.
Comment #1: Just an example of how the absence of technical specialists in the team of authors leads to an inappropriate design of the study: You reply "In the current study, we did not have access to manufacturer-proprietary specifications such as exact pulse duration, energy (nJ), or fluence (J/cm²) for both the VisuMax 500 and 800 platforms."But this is exactly what is not required. Because the laser power inevitably changes during the operation of the equipment. Instead, it is necessary to use a calibrated power meter to measure an average power and measure the size of the laser beam in the focal waist in order to calculate the fluence based on this information. I'm still recommend, that the technical specialists proofread the manuscript, to avoid possible errors.
Response 2: We thank the reviewer for this insightful comment and for highlighting the importance of involving technical specialists and employing broader methodological approaches. While we were unable to include an optical specialist as a co-author for this clinical study, we acknowledge the value of such collaboration and will carefully consider this in future work.
In our study, direct in-situ measurement of average laser power (?avg) and focal beam waist (?0) at the corneal stromal plane was not feasible, as the VisuMax 500 and 800 are sealed medical femtosecond laser systems with closed optical paths and patient-contact interfaces. Instead, we calculated the per-pulse fluence from the known pulse energy (??) and the programmed spot and track spacing (Δ? and Δ?) according to the equation:
This approach is commonly applied in raster-scanned femtosecond laser procedures, where each lattice cell receives approximately one pulse, and has been widely used for modeling photodisruption and corneal tissue cutting.
Under the standard clinical settings of the VisuMax 500 used at our institution during the study, the spot distance, track distance, and energy were 4.3 µm, 4.3 µm, and 125 nJ, re-spectively, resulting in a calculated fluence of 676 m J/cm². Under the standard clinical settings of the VisuMax 800, these parameters were 4.0 µm (spot distance), 3.0 µm (track distance), and 115 nJ (energy), yielding a calculated fluence of 958 m J/cm²
We have now added this calculation result in the Methods section, acknowledged it as a limitation, and noted that while these values do not incorporate direct beam profile measurements or pulse overlap factors, they fall within the range generally reported for smooth and complete femtosecond laser incisions.
[Methods, 2.4. Surgical Technique, LL 129-136]
Under the standard clinical settings of the VisuMax 500 used at our institution during the study, the spot distance, track distance, and energy were 4.3 µm, 4.3 µm, and 125 nJ, re-spectively, resulting in a calculated fluence of 676 m J/cm² using the equation of ? = ??/(Δ?⋅Δ?), where ?? is the pulse energy (J), Δ? is the spot distance (cm), and Δ? is the track distance (cm) [20]. Under the standard clinical settings of the VisuMax 800, these parameters were 4.0 µm (spot distance), 3.0 µm (track distance), and 115 nJ (energy), yielding a calculated fluence of 958 m J/cm² [20].
[Disscussion, LL 363-378]
Fourth, in this study, direct in-situ measurement of the average laser power (?avg) and focal beam waist (?0) at the corneal stromal plane was not feasible, as the VisuMax 500 and 800 are sealed medical femtosecond laser systems with closed optical paths and pa-tient-contact interfaces. Instead, we estimated the per-pulse fluence from the known pulse energy (??) and the programmed spot and track distances (Δ? and Δ?) according to the equation ? = ??/(Δ?⋅Δ?) [20]. This equation represents a standard approximation for ras-ter-scanned femtosecond incisions, where each lattice cell receives approximately one pulse. As a result, the calculated fluence values for the VisuMax 500 and 800 were similar. A previous study has indicated that a simple conclusion stating that smaller spot and track distances combined with higher laser energy result in minimal roughness cannot be drawn, as multiple interacting factors influence the resultant surface quality [20]. There-fore, based on the data obtained in this study, it remains unclear whether the differences in clinical outcomes are attributable primarily to laser–tissue interaction characteristics or to improved device features such as centration guidance and cyclotorsion compensation. Further studies comparing physical parameters across platforms may help clarify their respective contributions.
Comment #2: A serious, unbiased study, even if it focuses on observed clinical outcomes, should include an independent analysis of the effectiveness of devices manufactured by different companies. The difficulties in conducting such research are understandable - it often requires cooperation between several organizations. Nevertheless, the results of such studies can most likely be considered unbiased. In this work, this aspect requires significant attention. Conducting clinical trials directly comparing different devices from the same manufacturer is potentially unethical because:
Creates bias: Research inevitably focuses on the positive aspects of a given manufacturer's product line, presenting them in a favorable light.
It is a hidden advertisement: The results of such a study, even with an objective design promote the manufacturer's brand in the scientific literature, using trust in clinical data for marketing purposes.
Undermines scientific objectivity: The main purpose of clinical trials is an independent assessment of the benefits and risks for improving healthcare practices, rather than demonstrating the superiority of specific commercial products of one company.
Violates the principle of impartiality: It can create a conflict of interest, where scientific rigor is replaced by the task of highlighting differences within the brand, which does not serve the main purpose of choosing the best available treatment for patients.
I can only recommend inclusion of the equipment from other manufacturers into consideration, and adding transparent proofs, that the authors do not have conflict of the interests.
Response 2: We appreciate the reviewer’s important observation and fully agree that unbiased, multi-device evaluations are essential for scientific rigor. While the present study compares two devices from the same manufacturer, this selection was based solely on the availability of equipment at our institution and was not intended to promote any specific commercial product.
We acknowledge the potential for perceived bias inherent in such a design. To address this concern, we have added a statement to the Discussion section under the limitations paragraph, noting that the inclusion of devices from a single manufacturer may limit the generalizability of the findings. We have also emphasized the need for future multi-center studies involving equipment from multiple manufacturers to ensure more balanced and objective evaluations.
[Discussion, LL 357-363]
Third, this study compared two devices from the same manufacturer, which may introduce bias by unintentionally emphasizing the strengths of a specific product line. As a result, the findings may have limited generalizability. The devices were selected solely based on their availability at our institution, and the study was not intended for promotional purposes. Nonetheless, future research should include devices from multiple manufacturers, ideally through multi-center collaborations, to ensure more balanced and widely applicable evaluations.
Round 3
Reviewer 3 Report
Comments and Suggestions for Authors
Thank you for the reply. I sincerely ask you to excuse me, but I'm still not understand the difficulties with inviting the technical specialist for proofreading this paper and direct measurements the fluence "at the corneal stromal plane". Seoul have a lot of universities and scientific organizations, where such specialist can be easily found. For example, by publications through the databases like Scopus or Web of Science (you can use keywords like "ultrafast optics" or any other related to easily identify the field, that apply filter by the country). alternatively you may contact Seoul National University. The first random choice identified a group of researchers, who wrote this paper: https://doi.org/10.1109/JLT.2024.3454289. Power meters are standard tools in optical laboratories. In addition, a technician will allow you to avoid making blunders like you do over and over again by counting using a track to determine the fluence. Using the track created by a laser as a coordinate for estimating the area in fluence calculation is incorrect because the track represents the cumulative result of laser movement and interaction, not the true instantaneous beam spot size. Fluence requires knowing the beam's exact cross-sectional area at the target, not the potentially irregular or elongated track left by beam motion, which does not accurately reflect the energy distribution across the beam's actual interaction zone. And providing empty space (lines 467-468) instead of the reference [20] is also incorrect, so I even unable to check those literature source you refer. I still strongly recommend following my advice, it will be much more reliable and safer.
2. To speed up the discussion for at least one extra round, I also reformulate my question in another form: If you can place a patient's head with an eye which undergoing for the treatment at the laser beam path, why can't you place a simple optical scheme with a calibrated filter and a power meter in that place? In particular, please also reply, does your setup provides the possibility for control laser beam power? I think there should be such an opportunity.
3. Regarding the second comment: I'm glad, that you acknowledged the potential for perceived bias inherent in such a design of the study. But I believe, that your corrections may be not enough to ensure full transparency. If you refuse the proposal to collaborate with the other groups possessing another tools, please clearly disclose the potential for perceived bias in multiple sections of the manuscript, such as the abstract, introduction, methods, and discussion. Additionally, I recommend to compare and contrast your findings with studies using devices from different manufacturers. Highlight similarities and differences to provide context and acknowledge where results might diverge.
Finally, the editorial board should require authors to submit a comprehensive disclosure statement detailing any financial or personal interests related to the devices used in their study. This statement should outline any funding received, affiliations with manufacturers, and potential conflicts of interest. Authors should use clear and unambiguous language to specify their relationship with the device manufacturer and any financial benefits derived from the study. These disclosures should be prominently displayed within the published manuscript.
Author Response
1. Thank you for the reply. I sincerely ask you to excuse me, but I'm still not understand the difficulties with inviting the technical specialist for proofreading this paper and direct measurements the fluence "at the corneal stromal plane". Seoul have a lot of universities and scientific organizations, where such specialist can be easily found. For example, by publications through the databases like Scopus or Web of Science (you can use keywords like "ultrafast optics" or any other related to easily identify the field, that apply filter by the country). alternatively you may contact Seoul National University. The first random choice identified a group of researchers, who wrote this paper: https://doi.org/10.1109/JLT.2024.3454289. Power meters are standard tools in optical laboratories. In addition, a technician will allow you to avoid making blunders like you do over and over again by counting using a track to determine the fluence. Using the track created by a laser as a coordinate for estimating the area in fluence calculation is incorrect because the track represents the cumulative result of laser movement and interaction, not the true instantaneous beam spot size. Fluence requires knowing the beam's exact cross-sectional area at the target, not the potentially irregular or elongated track left by beam motion, which does not accurately reflect the energy distribution across the beam's actual interaction zone. And providing empty space (lines 467-468) instead of the reference [20] is also incorrect, so I even unable to check those literature source you refer. I still strongly recommend following my advice, it will be much more reliable and safer.
Response 1: We sincerely thank the reviewer for emphasizing the importance of accurate fluence assessment. We fully agree that direct fluence measurement at the corneal stromal plane would provide valuable additional data. However, our manuscript is based on a clinical comparative evaluation of the outcomes of two devices (VisuMax 500 and VisuMax 800). The focus of our work is on clinical results, not on benchtop measurements of laser beam characteristics. In addition, while we recognize that specialized laboratories in Seoul possess the equipment necessary to directly measure fluence at the stromal plane, it would not be feasible to invite independent researchers to transport such sensitive instrumentation to our hospital solely for the purpose of supplementing an already completed clinical study. Without prior planning or a shared research framework, such an arrangement would be both impractical and potentially inappropriate, as it would require substantial effort from other investigators without offering a meaningful collaborative benefit. Therefore, we respectfully emphasize that our approach is consistent with standard practice in clinical ophthalmic studies comparing refractive surgical platforms. To our knowledge, published clinical trials in this field typically evaluate outcomes using clinical endpoints, such as visual acuity, refractive predictability, and safety indices, without direct optical power measurements at the treatment site. We have followed this established convention, and our conclusions remain framed within the clinical scope of our study.
We apologize for the oversight regarding Reference 20. This has now been corrected to ensure that Reference 20 is visible.
In light of these considerations, we believe that incorporating direct fluence measurements lies beyond the scope of the present work. Nevertheless, we have revised the manuscript to make this limitation explicit in the manuscript, noting that we did not perform direct fluence measurements at the corneal stromal plane. We also highlight this as a potential direction for future research that combines clinical outcomes with benchtop optical analyses. We hope this clarification addresses the reviewer’s concerns, and we remain deeply grateful for the thoughtful suggestions aimed at strengthening the transparency and rigor of our work.
[References, LL 471-472]
- Fox T, Mücklich F. Development and validation of a calculation routine for the precise determination of pulse overlap and accumulated fluence in pulsed laser surface treatment. Advanced Engineering Materials 2023;25:2201021.
[Discussion, LL 367-382]
Fourth, this we did not perform direct measurements of laser fluence at the corneal stromal plane. As this was a clinical comparative study focusing on surgical outcomes of the VisuMax 500 and 800 platforms, such benchtop optical measurements were be-yond the scope of the protocol. Instead, we estimated the per-pulse fluence from the known pulse energy (??) and the programmed spot and track distances (Δ? and Δ?) according to the equation ? = ??/(Δ?⋅Δ?) [20, 28]. As a result, the calculated fluence values for the VisuMax 500 and 800 were similar. A previous study has indicated that a simple conclusion stating that smaller spot and track distances combined with high-er laser energy result in minimal roughness cannot be drawn, as multiple interacting factors influence the resultant surface quality [28]. Therefore, based on the data ob-tained in this study, it remains unclear whether the differences in clinical outcomes are attributable primarily to laser–tissue interaction characteristics or to improved de-vice features such as centration guidance and cyclotorsion compensation. Future in-vestigations integrating clinical outcomes with direct fluence measurements in optical laboratory settings may further elucidate the relationship between laser parameters and clinical results.
- To speed up the discussion for at least one extra round, I also reformulate my question in another form: If you can place a patient's head with an eye which undergoing for the treatment at the laser beam path, why can't you place a simple optical scheme with a calibrated filter and a power meter in that place? In particular, please also reply, does your setup provides the possibility for control laser beam power? I think there should be such an opportunity.
Response 2: We thank the reviewer for this important question. In principle, replacing the patient’s eye with an optical setup could allow direct power measurements. However, the systems include internal mechanisms to monitor and control laser power for safety, but these are proprietary and not open to external validation in a clinical setting. Our study was designed as a clinical comparative study, limited to clinical endpoints. Instead, as noted in our previous revision, we have provided fluence values calculated from the manufacturer’s device parameters, which represent the best available estimate within the approved clinical study framework. We now explicitly state this limitation in the revised manuscript and suggest that future collaborative studies combining clinical and laboratory analyses could address this important aspect.
- Regarding the second comment: I'm glad, that you acknowledged the potential for perceived bias inherent in such a design of the study. But I believe, that your corrections may be not enough to ensure full transparency. If you refuse the proposal to collaborate with the other groups possessing another tools, please clearly disclose the potential for perceived bias in multiple sections of the manuscript, such as the abstract, introduction, methods, and discussion. Additionally, I recommend to compare and contrast your findings with studies using devices from different manufacturers. Highlight similarities and differences to provide context and acknowledge where results might diverge.
Response 3: We appreciate the reviewer’s suggestion. Our clinical setup is not equipped to integrate additional optical hardware such as a calibrated power meter in the treatment beam path, since such modifications would interfere with regulatory approval and device integrity. Importantly, the study design focused on clinical outcomes under standard operating conditions of the commercially available VisuMax platforms. Nevertheless, we have clarified in the revised manuscript that we did not directly control or measure stromal plane fluence, and that our conclusions are based on clinical parameters only.
4. Finally, the editorial board should require authors to submit a comprehensive disclosure statement detailing any financial or personal interests related to the devices used in their study. This statement should outline any funding received, affiliations with manufacturers, and potential conflicts of interest. Authors should use clear and unambiguous language to specify their relationship with the device manufacturer and any financial benefits derived from the study. These disclosures should be prominently displayed within the published manuscript.
Response 4: We agree with the reviewer and have revised the manuscript accordingly. We now explicitly acknowledge the potential for perceived bias in the Abstract, Methods, and Discussion sections. We have prepared a comprehensive disclosure statement detailing our affiliations and any potential conflicts of interest; no external funding was received. We believe these revisions will enhance the transparency and rigor of the manuscript.
[Abstract, LL 28-29]
This study may be limited by perceived bias associated with evaluating two platforms from the same manufacturer.
[2. Materials and Methods, LL 80-83]
This investigator-initiated study received no financial support from the device manufacturer. As both devices were from the same company, we acknowledge the potential for perceived bias and address it by comprehensively and transparently reporting outcomes.
[Discussion, LL 361-365]
Third, this study compared two devices from the same manufacturer, which may introduce bias by unintentionally emphasizing the strengths of a specific product line. As a result, the findings may have limited generalizability. The devices were selected solely based on their availability at our institution, and the study was not intended for promotional purposes.